# Case Report: Emerging Losses of Managed Honey Bee Colonies

**DOI:** 10.3390/biology13020117

**Published:** 2024-02-13

**Authors:** Zachary S. Lamas, Yanping Chen, Jay D. Evans

**Affiliations:** USDA-ARS Bee Research Lab, BARC-East Bldg. 306, Beltsville, MD 20705, USA; judy.chen@usda.gov

**Keywords:** honey bees, collapse, agricultural pests, pathogens, parasites, economic harm

## Abstract

**Simple Summary:**

*Apis mellifera* is a managed pollinator that experiences cycles of extreme losses at the population level. Several culprits have been associated with these mass losses. Most notably, the parasitic mite *Varroa destructor* is a consistent suspect in periodic mass losses of honey bee colonies. Our team performed diagnostic screening of pathogens and parasites in operations that experienced extreme losses during the winter of 2023. The direct cost to replace lost colonies and the immediate economic harm imparted from severe losses are described.

**Abstract:**

United States commercial beekeepers prepare honey bee colonies for almond pollination in California each year in late January to early February. This represents the largest managed pollination event in the world and involves more than half of all U.S. honey bee colonies. In winter 2023, numerous colonies in Florida, which were graded as suitable for almonds (larger than ten frames of bees), dwindled suddenly or altogether died within several weeks, just prior to movement for almonds. The timing of these losses and the resulting morbidity caused severe economic harm to affected operations. This study reports interviews with affected stakeholders, their economic harm, and analyses of pathogens and parasites found in their colonies.

## 1. Introduction

Managed honey bees *(Apis mellifera)* are an integral component of North American agriculture by providing pollination services to fruit, vegetable, oil and nut crops. However, honey bees are under threat from a series of stressors that affect their health [1]. These stressors of colony loss are responsible for historically high yearly losses in managed honey bee populations [2]. The stabilization of managed bee populations has only been possible through intense splitting; the act of reproducing colonies through artificial division of healthy surviving colonies [3]. However, beekeepers are especially vulnerable when yearly losses leave them with few resources to replace deceased colonies. When this happens, beekeepers may have to purchase new colonies or forgo pollination events, their largest source of income.

Although the stressors which cause colony loss are well described, the economic cost of colony replacement and the economic harm associated with immediate loss in pollination income imparted by those stressors are not. In colony collapse events, the economic harm is assumed to be severe on commercial operators. Here, we provide a case report of a collapse event in the winter of 2023 amongst commercial operations in Florida, USA. This case study includes interviews detailing emotional stress imparted by the losses, financial harm, and pathogen and parasite screenings of surviving colonies.

## 2. Methods

We used targeted sampling methodology to build a network of beekeepers experiencing similar symptoms and losses in the state of Florida [4]. Information from these recruits was used as *a priori* information directing how to sample colonies.

No yard was small enough to inspect, let alone sample in its entirety, in a reasonable amount of time. As a result, random sampling was not performed. Instead, symptomatic colonies were marked during a brief walk through the apiary. A random sample of these symptomatic colonies was then sampled. Adjacent asymptomatic colonies were also sampled. One sampling event was shortened due to extreme stinging within the yard.

### 2.1. Identifying Symptomatic Colonies

Symptomatic colonies were first identified visually by the presence of crawling bees in front of the colony or a pile of recently deceased bees. A colony would also be labeled symptomatic if, upon opening, morbidity in adult bees was observed. Since beekeepers complained about loss of adult bees, the decision to categorize a colony as symptomatic/asymptomatic was based on the state of adult bees. We chose to not use overt pathology of deformed wing virus (DWV) in workers to categorize symptomatic colonies at the time of inspection, though the observations were fully noted.

### 2.2. Sample Collection

A 50mL centrifuge tube containing approximately 100 adult bees was collected by shaking a frame of bees collected from the center of the brood nest. A standard alcohol wash was performed in order to sample *Varroa* from adult bees [5]. In short, a ½ cup of adult bees was collected in isopropyl alcohol from the same frame.

### 2.3. Qualitative and Grading Assessments of a Colony

A brood frame was removed, set in front of the colony and photographed. If the colony had dwindled, we used a hand in the photo for scale to describe the cluster. The number of frames of broods was recorded. The number of frames of bees was also recorded. Queen status was recorded.

Newly emerged bees were individually picked up and checked for mites. The total number of newly emerged bees with and without mites was recorded, and samples of both (with mites) were recorded. Up to 40 newly emerged bees per colony were inspected.

Irregularities in larval form, queen status and wax cappings were recorded. If cells were uncapped, then the pupa was removed, and a light shone inside the cell. If mites were observed, it was noted. Colonies were graded as strong, medium or weak.

If a colony had collapsed or looked like a quintessential CCD colony, it was recorded as such.

### 2.4. Molecular Analysis

Colonies were screened for pathogens using standard methods [6]. In short, a 50-bee sample was extracted for total RNA through a trizol, chloroform extraction and then converted to cDNA using BioRad Iscript prior to qPCR.

### 2.5. Reporting Survey Results

Private, personal information was provided during interviews with stakeholders. Raw data are not provided to respect the anonymity of operations, which shared sensitive, personal, information during these interviews. The size, location and production of an operation can provide enough information to single out individuals. As a result, we made two groups when presenting our data to maintain anonymity. The nine commercial pollinators were grouped together, and then data from their interviews were supplied together. To protect the anonymity of the queen breeders and honey producers in this study, their survey results were grouped together. When reporting personal information, we used the term “multiple operators” or “other operator”, which should be inferred as at least one.

### 2.6. Statistical Analysis

All data were analyzed using base R, in R Studio. Associations of target pathogens were analyzed using a Spearman coefficient, using *p* < 0.05 for statistical significance. Associations between parasite infestation and colony strength were analyzed using the non-parametric Kruskal–Wallis test. The Pearson coefficient was used for associations between pathogen load and size of colonies. 

## 3. Case Report

In January 2023, beekeepers in the state of Florida began reporting unexpected losses of their managed honey bee colonies and, simultaneously, an unusually large number of weak colonies. The reports came from two separate groups of beekeepers. The first report was from commercial queen producers in South Florida who sent videos of adult bees exhibiting morbidity. These queen breeders know each other but keep bees separately. They experienced unexpected whole colony losses and an almost complete loss of their forager bees (older bees comprising up to half of the colony). The second wave of reporting occurred several weeks later from a larger operator in Northern Florida, who reported similar symptoms. We created a network of affected operators using targeted sampling “snowball” methodology by encouraging initial reporters to self-recruit amongst their colleagues to assess losses across the network of commercial operators. We quickly determined through interviews that reported losses and morbidity were unusually high and were causing severe economic harm to affected operators.

Information gathered from interviews was used *a priori* to direct sampling protocols for an immediate sampling expedition. Since beekeepers reported sudden losses of their adult bee populations, our sampling focused on the collection of adult bees. Because survivorship bias may occur, efforts were made to locate colonies which still had symptomatic bees, and to capture those bees alongside their asymptomatic nestmates.

Upon arrival, we found operators across the state with thousands of unexpectedly weak colonies in their commercial apiaries (Figure 1). These operators had inspected and graded their colonies for the upcoming almond pollination season in California. These colonies must have a minimum of 10 frames of bees and 6 frames of brood (developing bees) to meet the required grade, with many operators sending colonies with upwards of 20 frames of bees. When beekeepers returned to their colonies in January, they found thousands of those colonies had suddenly dwindled to two to three frames of brood over the course of three weeks (Figure 1).

Beekeepers self-reported mosquito spraying as a possible culprit for the losses. There was an increase in mosquito spraying after Hurricane Ian, and every county in which bees were located had sprayed a combination of larvicide and adulticide sprays through a combination of ground and aerial dispersal. Nevertheless, the most persistent observation in surviving colonies was the presence of *Varroa destructor*, a parasitic mite that causes direct damage to bees by feeding on their fat bodies and hemolymph and indirect damage by vectoring viruses when they feed on adult bees and developing broods [7]. Despite repeated treatments, *Varroa* was detected in colonies across the study (71.6% of colonies with at least one detection). Alcohol washes were performed on each colony in the study and varied greatly around the mean (M = 2.57%, SD = 3.88%). There was no significant difference in mite levels between colonies of different strengths (H(2) = 0.0298, *p* = 0.985). Figure 2 shows the percentage of mites detected in adult bees. Beekeepers reported applying miticides multiple times in October and November, with follow-up treatments reported by most operators in January. However, clinical signs of mite infestation were observable in 44.6% of the colonies sampled (N = 74). Additionally, many healthy-appearing asymptomatic colonies, not showing any signs of disease or typical mite damage, harbored infestations of *Varroa* higher than economic thresholds [8].

The alcohol wash was not deemed a universally reliable method to report what *Varroa* infestations might have recently been since beekeepers had used repeated treating cycles. For example, two operations reported treating up to two weeks prior to our sampling. Those colonies reported very low levels of *Varroa,* even though clinical signs of high mite levels were visible in the sampled colonies. Consequently, other methods to detect Varroa were implemented. Namely, worker bees were inspected at emergence for *Varroa.* Some colonies had a high level of infestation in their worker brood (0–90%, M = 15.78%, SD = 23.43%).

### 3.1. Pathogens in Sampled Colonies

Pathogen screening measured the levels of key viruses, the microsporidian Nosema and trypanosome parasites that are known to affect honey bee health and have been associated with historical bee losses. All colonies had at least one pathogen detected, with the majority of colonies having multiple detections (95.8%). The most prevalent pathogen detected across all colonies was *Nosema ceranae*, followed by deformed wing virus (DWVA), and then black queen cell virus (BQCV). Other pathogens that were detected were detected at low frequencies (Table 1). Israel acute paralysis virus (IAPV), a pathogen previously associated with colony collapse [9], was not detected in any of the sampled colonies. Another paralysis virus, Kashmir bee virus (KBV), was also not detected, while chronic bee paralysis virus (CBPV) was rarely detected. However, acute bee paralysis virus (ABPV), a paralysis virus associated with *Varroa destructor,* was present in 20.8% of sampled colonies.

There was no significant difference in viral levels between colonies of different strengths (Table 1), nor was there a strong correlation between colony size and viral levels. Several pathogens were associated with each other. DWVA was associated with BQCV, DWVB, ABPV and *N. ceranae*. These relationships were positive, moderate in strength and significant (Table 2). ABPV had a similar association with *N. ceranae*, as did *N. ceranae* with trypanosomes. Levels of screened pathogens are depicted in Figure 3.

There was a significant, positive correlation between the percentage of mites found in emerging workers, or the alcohol wash, and key pathogens that affect honey bee health (Table 3).

### 3.2. Economic Harm

Twelve beekeeping operations were interviewed and sampled in this study. Of those operations, nine were commercial operators who regularly provided pollination services. Two remaining operations commercially produced queens (2), while one was a non-migratory honey producer within the state of Florida.

Each operation responded to an economic harm survey, which was designed to uncover the immediate loss of income incurred by these events and the cost to recover. Beekeepers were asked to only include immediate loss of income and not to include extended losses beyond March 2023. When including costs associated with extreme morbidity, operations were asked to report the cost to fix or replace these dwindled colonies. Several operators lost 90% or more of their anticipated income when colonies either died suddenly or dwindled to economically unviable clusters.

Outright loss of colonies from November 2022 to February 2023 was estimated through interviews with operations. Colony losses due to Hurricane Ian were excluded from these estimates. Losses between November 2022 and the beginning of February 2023 ranged from 20% to 88.9% (M = 48.1%, SD = 22.7%). This represented a total of 20,567 colonies lost across 12 operations in a span of approximately three months (48.6% loss). An estimated 21,671 colonies remained, with 6727 (31.5%) of these colonies being economically unviable units. These dwindled colonies could not provide any service within their operations as they did not meet the minimum requirements for pollination contracts, queen production or honey production. The dwindled colonies would not be able to provide surplus resources to make new colonies with either. 

The responses from beekeepers varied greatly with respect to how they would manage these dwindled colonies. These responses were reflective of the circumstances and resources unique to each operation. Two of the smaller operators believed keeping their dwindled colonies alive posed a greater risk to their healthy colonies than the cost to euthanize them. Consequently, they responded by euthanizing their dwindled colonies. As a result, their cost to fix those colonies was relatively low (USD 20 per colony). On the other hand, the larger operators allowed the dwindled colonies to persist in their holding yards. Beekeepers will repair or replace these dwindled colonies, potentially introducing new pathogens into their operation by the large-scale purchase of bees.

When asked how they would fix or replace the dwindled colonies, answers varied. Three operators said they would or were considering purchasing new colonies to replace their dwindled colonies. The remaining beekeepers would fix the colonies using resources within their own operations. Two operators responded they had neither the financial resources to replace nor the bee resources within their operations to fix their dwindled colonies. Dwindled colonies represented 44% to 100% of the remaining colonies of these operations. The total cost to replace or fix the 6818 dwindled colonies was estimated to be USD 908,630, based upon interviews with each operation.

### 3.3. Loss of Income from Commercial Pollination

Severe losses and morbidity resulted in the immediate loss of pollination income for 9 of the 12 operations in this study. Operations sent 34.1% to 100% fewer colonies to almonds than anticipated (M = 70.4%, SD = 22.9%). This represented an estimated loss of USD 3,451,420 anticipated income from almond pollination across the nine operations (Figure 4).

### 3.4. Loss of Income from Queen and Honey Production

The three other operators produced honey and or queens instead of engaging in commercial pollination services. We surveyed for their immediate loss of income by covering anticipated production during early spring and arrived at an estimated loss of USD 825,700. We combined the estimates from all three operations purposefully to maintain the anonymity of the beekeeping operations.

### 3.5. Loss of Employment

Five of the twelve operations faced layoffs (42%), resulting in a total of eight jobs lost.

## 4. Discussion

The commercial beekeeping operations in this study reached out for help due to both severe economic harm and severe loss and morbidity of their managed bees. Symptoms and losses were reflective of national losses experienced in 2007–2009 [9,10,11]. Information garnered through beekeeper interviews directed the field collection techniques. From information acquired from interviews and laboratory analysis, pathogens and mites and/or exposure to mosquito spraying were deemed the most likely culprits. At the time of our initial field response, we believed we were responding to severe losses acute to beekeepers in the state of Florida. However, since that time, unusually high losses and similar symptoms have been reported from multiple states. Such a broad geographic area for losses reduces the likelihood that pesticides are a key culprit.

We did not find a significant difference in viral loads or mite levels between colonies of different strengths. This may be because *Varroa* and viral infections can be asymptomatic or symptomatic in honey bee colonies.

The nine commercial pollinators were all treated for *Varroa* once their colonies returned to Florida in the fall of 2023. Each operation used multiple applications of a miticide as a single treatment. This was to encompass an entire brood cycle, ensuring mites were routinely in contact with the miticide as they emerged from brood cells. All of these operators used amitraz followed-up with an oxalic acid treatment, both of which are standard treatments for beekeepers. One operation used thymol as a follow-up treatment. Two of the operations knew they had high mite levels following treatments and treated two and five weeks prior to our sampling in early February. At least one operator showed *Varroa* levels were high in August before transporting colonies back to Florida. Despite repeated treatments, high mite levels were observed in multiple operations in the study. Of the 74 colonies sampled, 15 of them (20.3%) were above the economic threshold of 3%. Persistent, high levels of infestation like this create concern over the health of operations going forward as infested colonies can re-infest neighboring colonies through the dispersal of mites while simultaneously promoting ongoing viral circulation within operations [12,13].

Multiple stressors are known to impact honey bee health. In addition to viruses, *Nosema ceranae*, a unicellular parasite, was the single most prevalent pathogen in the case study. Detected at high levels in many colonies, *N. ceranae* has been previously implicated in sudden colony losses [14].

Severe bee losses have garnered much public attention in the past, resulting in media frenzies and speculations about possible culprits causing the historical losses. Unfortunately, these frenzies do a disservice to the commercial beekeepers who struggle to keep colonies alive for commercial honey production and pollination needs. We highlight that these events impart severe economic distress onto small- and medium-sized agricultural businesses. The beekeepers interviewed in this study are impacted financially, emotionally and mentally. The struggles these beekeeping operations face mirror the economic frailty, volatility and stress other family farm operations deal with [15]. Multiple operators were going to pull from their retirement funds to continue their operations. Multiple operators are near retirement and would not keep beekeeping had they not run into recent issues. They said they need to have one more great year and the ability to sell off their colonies to afford retirement. In short, these beekeepers are playing roulette with their retirement funds. Other operators have not told their spouses of the severe losses yet, and instead go to work each day counting the lost colonies and stacking empty equipment, their spouses still thinking they are leaving home each day to take care of their beloved bees. Other operators are seeking intra-family loans to recover by asking an off-the-farm family member for money. We describe these personal stories to underscore the vital role family farms play in food production in the United States. Events which cause severe economic harm to these operations have a tangible, human element and affect food security.

The causes of these bee losses are still unknown. The symptoms are iconic of colonies which collapse due to heavy infestations of *Varroa destructor.* Beekeepers largely began treating their colonies in October and November 2022. If their initial treatments were not effective, and mite levels remained high, then significant damage could have been imparted onto their worker bee populations before the mite levels were corrected. Beekeepers have been reporting ineffectiveness of a widely used miticide, Amitraz, and there is evidence of genetic resistance developing in mite populations [16]. Further studies into mite resistance to Amitraz are imperative as commercial beekeeping is reliant upon this miticide [17].

A change in the virome of honey bees could also be a likely culprit. RNA viruses replicate at high frequencies and readily make new variants [18]. An emerging issue such as a change in the virome or resistance to Amitraz would likely be unequally distributed across honey bee populations at this time, and we would see collapses within select operations, as we did this winter.

Several culprits have been suggested, which would not be emerging in nature. Forage was reported as poor in several parts of the nation in 2022. Florida, in particular, lost a reliable nectar and pollen source; Brazilian Pepper, *Schinus terebinthifolius* Raddi, was lost in the late fall due to Hurricane Ian. Beekeepers self-reported mosquito spraying and, in particular, larvicidal spraying, as a likely culprit. There is correlative support for this claim, as mosquito spraying had increased in some areas due to Hurricane Ian, and every county which housed bees in this study sprayed both adulticides and larvicides. These two culprits could have acted synergistically, though neither can explain similar losses being reported now across a much larger geographic range having since been reported.

Whatever the culprit or culprits, beekeepers have been severely affected by these loss events. It is of the utmost importance for the commercial beekeepers participating in this study to pinpoint the underlying issue before the arrival of winter in 2024. Many beekeepers said they could not recoup from a similar event if it were to happen in 2025. For multiple beekeepers, the winter of 2023 followed a disappointing winter in 2022, where they had already sustained economically strenuous losses.

## 5. Conclusions

Similar reports of severe losses and of dwindled, non-productive colonies, which refuse to grow, have since been reported in early 2023 from Utah, California, Oregon and from numerous indoor storage shelters across the Western United States. The broad and growing geographic range in which colony losses are reported is suggestive of an emerging issue in managed honey bee colonies within the United States, arguably related to contagious disease. A concerted effort between stakeholders and researchers is underway to identify potential culprits. Previous periods of high colony loss, like in 2007, when Colony Collapse Disorder (CCD) was reported, led to a media frenzy and a flurry of speculative reports of the potential culprits causing colony collapse [11]. Ongoing engagement with commercial operators and multiple research groups is continuing to determine if this is an emerging issue in honey bee colonies within the United States and to collect critical data for identifying the generalities of bee declines.

Despite the uncertainty over potential culprits for colony losses, the economic harm those losses pose is certain. Numerous operations lost a majority, or all, of their immediate income while being left with ongoing operation expenses and the task of rebuilding. Simply put, these events pose a risk to the livelihood of commercial beekeepers. Identifying the causes for colony collapse is tantamount to provide reliable, large-scale pollination services and a thriving commercial beekeeping industry.

**Author contributions:** Z.S.L. collected samples, analyzed colonies, and carried out stakeholder interviews. Z.S.L. processed samples. Y.C. and J.D.E. supervised sample processing. Z.S.L., Y.C. and J.D.E. analyzed the data. Z.S.L. wrote the manuscript. Y.C. and J.D.E. edited the manuscript. All authors have read and agreed to the published version of the manuscript.

## Figures and Tables

**Figure 1 biology-13-00117-f001:**
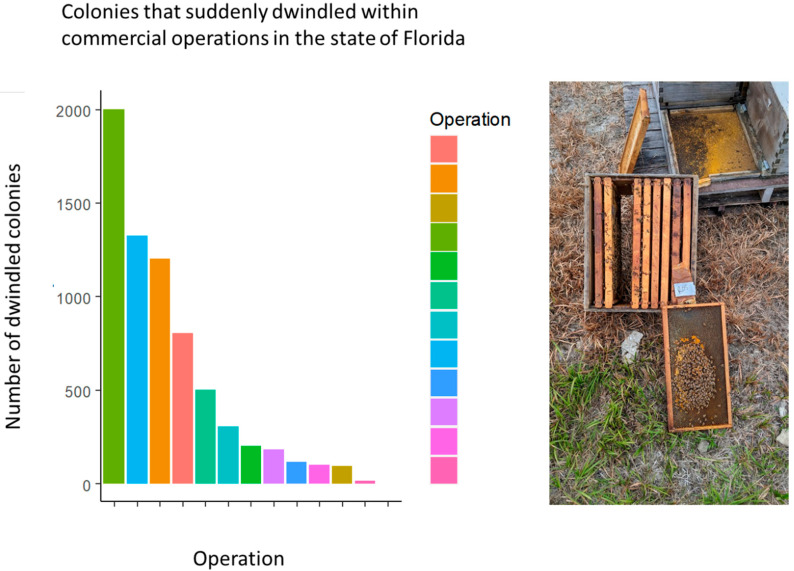
**(Left)** The number of colonies in each operation that suddenly dwindled. (**Right**) A typical dwindled colony with a patch of brood and insufficient adult bees to cover the brood area. This colony lost approximately 90% of its adult bee population in the span of 3 weeks.

**Figure 2 biology-13-00117-f002:**
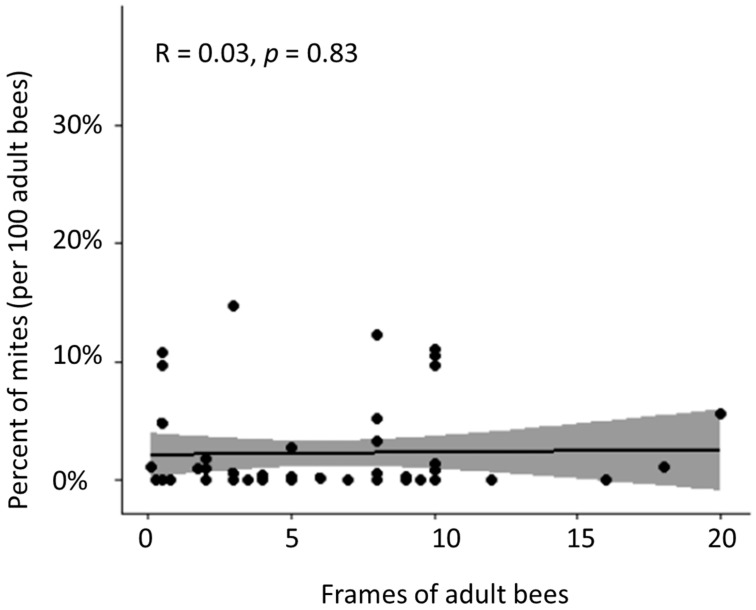
*Varroa destructor* detections on adult bees in colonies of various sizes (r(51) = 0.03, *p =* 0.83). Dots represent individual colonies, shown with regression fit line and 95% confidence band.

**Figure 3 biology-13-00117-f003:**
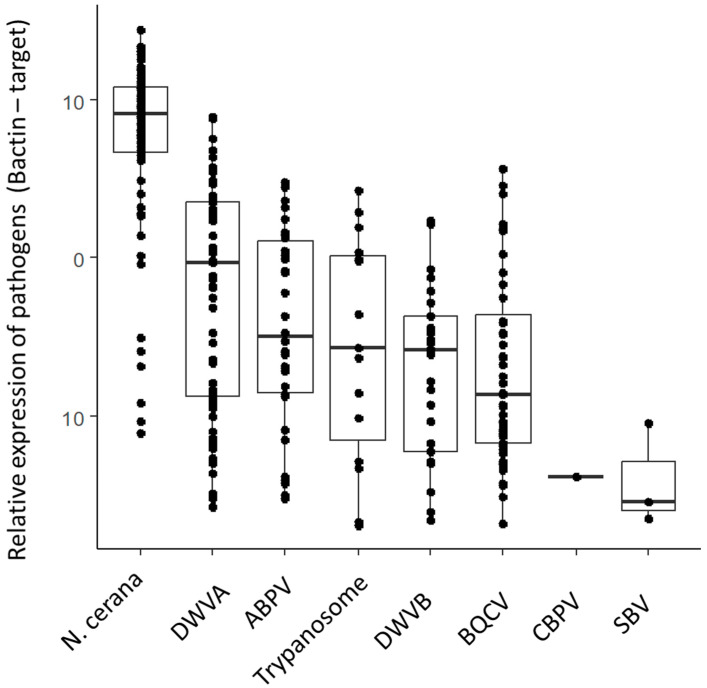
Pathogen targets in sampled colonies, box plots showing 95% confidence interval, line showing standard deviation.

**Figure 4 biology-13-00117-f004:**
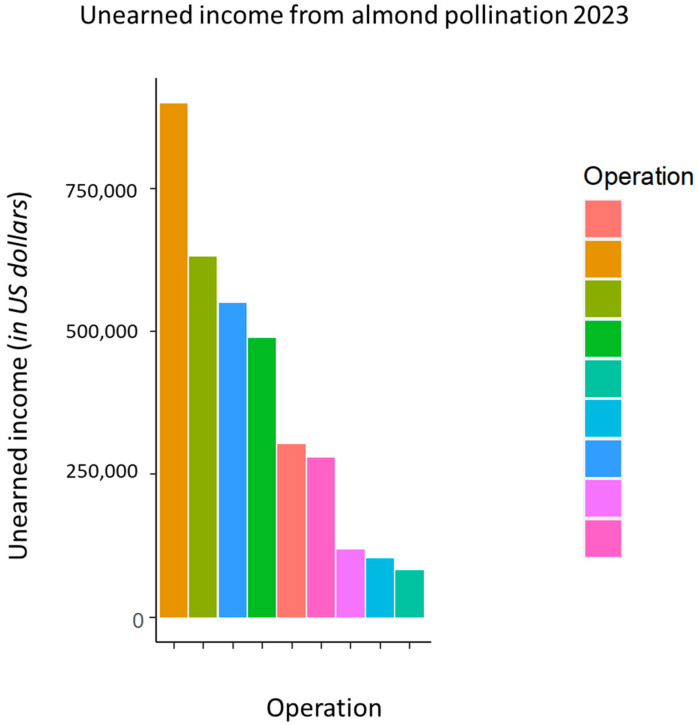
Of the 12 beekeepers surveyed in this study, 9 were commercial migratory beekeepers who performed pollination services. These operations were expected to bring thousands of colonies to California to fulfill pollination contracts. Operations missed out on an estimated USD 3,451,420 of income due to the severe losses and morbidity in late January.

**Table 1 biology-13-00117-t001:** Prevalence of 10 pathogen targets in colonies of different strengths from Florida.

Prevalence of Pathogens in Sampled Colonies										
Colony Strength	DWV	DWVB	ABPV	BQCV	CBPV	IAPV	KBV	SBV	*N. ceranae*	Tryp
Strong	1	0.385	0.231	0.692	0	0	0	0	1	0.615
Medium	0.864	0.409	0.182	0.864	0	0	0	0.045	1	0.5
Weak	0.87	0.46	0.216	0.76	0.027	0	0	0.054	0.972	0.405
All Colonies	0.889	0.431	0.208	0.778	0.014	0	0	0.042	0.986	0.472

**Table 2 biology-13-00117-t002:** Associations of target pathogens (using Spearman coefficient, df = 70, insignificant associations are printed in normal typeset, bold *p <* 0.05, bold * *p <* 0.001).

	DWVA	DWVB	BQCV	ABPV	*N. ceranae*	Tryp
DWVA		**0.42 ***	**0.26**	**0.36 ***	**0.37 ***	0.22
DWVB			0.041	0.15	0.082	0.11
BQCV				0.009	0.17	−0.026
ABPV					**0.32**	0.093
*N. ceranae*						**0.36**

**Table 3 biology-13-00117-t003:** Associations of target pathogens and infestation levels of adult bees or emerging brood (using Spearman coefficient, df = 70, insignificant associations are printed in normal typeset, bold *p <* 0.05, bold * *p <* 0.001).

	DWVA	DWVB	BQCV	ABPV	*N. cerana*	Tryp
Alcohol wash	**0.46 ***	**0.32**	0.19	**0.27**	0.18	**0.33**
Percent of emerging workers with mites	**0.66 ***	**0.61 ***	**0.38**	0.24	**0.35**	0.28

## Data Availability

Data will be made available in a public repository.

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
