# Peer review of "Case Report: Emerging Losses of Managed Honey Bee Colonies"

_biology, 2024, doi:10.3390/biology13020117_

Round 1

Reviewer 1 Report

Comments and Suggestions for Authors

 I consider the Case Report: “Emerging losses of managed honey bee colonies” should be accepted with minor revisions.

The monitoring and attention of the cases sheds light on a still unresolved problem of mortality of productive colonies in key regions of the United States.

This report allows us to visualize the problem and the implications that come with the loss not only of bees but also of economic resources, which is why many beekeepers see their income and their ability to continue in the activity affected or sacrifice other resources to recover what was lost. Making visible the impact that these colony losses have at the level of family economies and pollination services can help deepen and expand the case studies to find more concrete answers.

It is important to highlight that these losses are probably better explained from the point of view of sanitary-productive management (control of varroa, nosema, etc.) and the lack of resources in the environment , but resistance and mutation studies of circulating viruses should not be ignored.

It would be interesting if researchers could continue with these monitoring and surveys in the coming years, expanding the cases surveyed.

I highlight the contributions provided to the scientific community and accept this manuscript with minor corrections that must be made to continue with the publication process.

Brief suggestions for improving your manuscript:

line 59: cite the bibliographic reference of the type of varroa sampling carried out.

lines 63: improve the explanation of the sampling carried out. It is not clear.

Lines 51, 59 and 63: it would be more illustrative if the authors incorporated images.

Line 93: All data was analyzed using R and R Studio...

line 89 to 91: delete the paragraph since it is not appropriate to place it here. belongs to the summary.

figure 1: it could be interesting to discriminate between the three classes of operation.

lines 163, 174, table 2 and 3, Figure 3: correct DWVA by DWV.

line 262: this information was mentioned previously.

line 273: add "4". Discussion and Conclusions

Author Response

I consider the Case Report: “Emerging losses of managed honey bee colonies” should be accepted with minor revisions.

The monitoring and attention of the cases sheds light on a still unresolved problem of mortality of productive colonies in key regions of the United States.

This report allows us to visualize the problem and the implications that come with the loss not only of bees but also of economic resources, which is why many beekeepers see their income and their ability to continue in the activity affected or sacrifice other resources to recover what was lost. Making visible the impact that these colony losses have at the level of family economies and pollination services can help deepen and expand the case studies to find more concrete answers.

It is important to highlight that these losses are probably better explained from the point of view of sanitary-productive management (control of varroa, nosema, etc.) and the lack of resources in the environment , but resistance and mutation studies of circulating viruses should not be ignored.

It would be interesting if researchers could continue with these monitoring and surveys in the coming years, expanding the cases surveyed.

I highlight the contributions provided to the scientific community and accept this manuscript with minor corrections that must be made to continue with the publication process.

Thank you for this positive assessment of our work and its potential impact for researchers and stakeholders alike.

Brief suggestions for improving your manuscript:

line 59: cite the bibliographic reference of the type of varroa sampling carried out.

We agree a citation is needed for this. However, we placed it in methods. Hope this is ok. We placed the sentence with citation, “A standard alcohol wash was collected from each colony[9].” At line 238

lines 63: improve the explanation of the sampling carried out. It is not clear.

I amended this by stating we did a standard alcohol wash, and then gave a citation for the method.

Lines 51, 59 and 63: it would be more illustrative if the authors incorporated images.

We are unsure how we could apply images to those lines. We have separately added images which support statements in this article including hive conditions. We believe reviewers have been able to see those colony pictures.

Line 93: All data was analyzed using R and R Studio...

Oddly these lines were removed from the revised document. I pasted them back in, and slightly changed the highlighted text to using base R, in R studio.

line 89 to 91: delete the paragraph since it is not appropriate to place it here. belongs to the summary.

We respectfully disagree. Lines 89 – 91 are methods and belong in the methods section. The first clause of that paragraph could be placed in the discussion/conclusion, but coupled with the remaining two sentences it clearly tells other investigators how data was collected and presented.

figure 1: it could be interesting to discriminate between the three classes of operation.

We thought of that but unfortunately we do not have enough operations represented in each class to make this comparison in this study. Additionally, because we have so few we risk “outing” the identification of a single operator.

lines 163, 174, table 2 and 3, Figure 3: correct DWVA by DWV.

Respectfully we will leave the lines and the figure as is. We do mean to differentiate between DWVA and DWVB variants of deformed wing virus. Unfortunately, there is still no consensus on the exact vernacular to describe these two variants. Some researchers still use VDV to refer to DWVB. We recently published in Plos Pathogens and describe DWVA and DWVB and would like to keep the same formatting here for consistency across publications and ease of understanding for your readership.

line 262: this information was mentioned previously.

Unfortunately, some of the line numbers being requested to be fixed don’t align with the revision doc. I apologize in advance if we are inadvertently referring to different lines within the text. I believe the repetitive line is Colonies which dwindled to economically unviable clusters will be replaced or fixed through normal beekeeping practices by operators.

The text was changed by combining sentences to, “Operators estimated that replacement or fixing of colonies which dwindled to economically unviable clusters would cost $875,960,”. This removed the redundancy yet kept clarity about which colonies were causing what economic harm.

line 273: add "4". Discussion and ConclusionsThis was done. Thank you

Reviewer 2 Report

Comments and Suggestions for Authors

I believe that the manuscript has relevant information that should be published, mainly regarding the prevalence of viruses and other diseases. However, the document has many errors that must be corrected so that it can be published. Below I expose some of the errors. detected in the manuscript: N. ceranae and non-cerana, use of italics in scientific names, in line 160 there are missing references that support this comment, the use of the letter M as a symbol for mean is incorrect. I don't think the content of lines 262 to 272 adds value to the document. Furthermore, I suggest improving the discussion and adding more references regarding the prevalence of viruses and varroasis and their relationship with the loss of colonies.

Author Response

Rev 2

I  believe that the manuscript has relevant information that should be published, mainly regarding the prevalence of viruses and other diseases. However, the document has many errors that must be corrected so that it can be published.

Below I expose some of the errors. detected in the manuscript: N. ceranae and non-cerana, use of italics in scientific names, in line 160 there are missing references that support this comment, the use of the letter M as a symbol for mean is incorrect.

I don't think the content of lines 262 to 272 adds value to the document.  

We respectfully do not agree with this assessment. Lines 262-272 in the discussion report the economic harm incurred by these operators. As far as we know this is the first assessment of harm connected to a severe, colony loss event. This data can be used by researchers, stakeholders and policy makers alike to create financial risk assessments.

Furthermore, I suggest improving the discussion and adding more references regarding the prevalence of viruses and varroasis and their relationship with the loss of colonies.

 Respectfully, this is a case report and we want to focus on the traits that differentiate the tested symptomatic and asymptomatic colonies. We therefore report findings from our interviews, observations, and screenings, and palce these in the context of experimental evidence for causes of bee losses. However, we did not prove that Varroa and viruses caused the collapses here. We have strong evidence, similar to other crashes, that they are implicated and likely responsible. However, we can not exclude other confounders. We risk misdirecting other researchers and stakeholders by making too strong of statements about Varroa/viruses, and that includes lengthening the discussion to talk about varroasis/viruses. This topic has also recently been covered in several papers and reviews.

Reviewer 3 Report

Comments and Suggestions for Authors

Apis mellifera is a managed pollinator that experiences cycles of extreme losses on the population level. Several culprits have been associated with these mass losses. Most notably, the parasitic mite Varroa destructor is a consistent suspect in periodic mass losses of honey bee colonies.

This is an interesting study that reports interviews with affected stakeholders, their economic harm, and analyses of pathogens and parasites found in their colonies.

Following the introductory section of the article, the case report is well presented and details the events of colonies dwindling which have been reported through direct interview.

This type of report is useful to draw attention to these events of colony dwindling stimulating their investigation through appropriate epidemiological investigations, laboratory analysis directed to both parasites and pathogens as well as pesticide application.

Minor changes:

Line 61: Please give a more precise amount of bees collected, approx. 100 bees

Line 145: reference needed

Line 170: Table N. ceranae replace with N. ceranae

Line 179: Table 2 as above

Line 180: Figure 3 as above

Line 187: Table 3 as above

Line 352: Raddi should not be in italics

Author Response

Rev 3

Apis mellifera is a managed pollinator that experiences cycles of extreme losses on the population level. Several culprits have been associated with these mass losses. Most notably, the parasitic mite Varroa destructor is a consistent suspect in periodic mass losses of honey bee colonies.

This is an interesting study that reports interviews with affected stakeholders, their economic harm, and analyses of pathogens and parasites found in their colonies.

Following the introductory section of the article, the case report is well presented and details the events of colonies dwindling which have been reported through direct interview.

 Thank you

This type of report is useful to draw attention to these events of colony dwindling stimulating their investigation through appropriate epidemiological investigations, laboratory analysis directed to both parasites and pathogens as well as pesticide application.

Thank you, we are glad that assessment was understood. We have evidence of parasite/pathogen culpability, but we did not want to misdirect from pesticides. We are glad that was understood.

Minor changes:

We clarify now in the text that one centrifuge tube contains around 100 bees.

Line 145: reference needed

I added Delaplane, 1999 as a reference to line 145

Line 170: Table N. ceranae replace with N. ceranae

Thank you, I corrected this.

Line 179: Table 2 as above

Thank you, I corrected this.

Line 180: Figure 3 as above

Thank you, I corrected this.

Line 187: Table 3 as above

 Thank you, I corrected this.

Line 352: Raddi should not be in italics

 Thank you, I corrected this.

Round 2

Reviewer 2 Report

Comments and Suggestions for Authors

Appreciable authors

I value your work, I consider that your results are important for beekeeping. However, I believe that the presentation of the document should be improved. Below I list some suggestions.

Line 59. The titles should follow the same format, this title is posed as a question although it does not have the question mark. I suggest just putting: sample collection

Line 90. The title does not have the numbering. Add or remove all subtitles

Line 95. Add corresponding numbering, Start with a period

170-171. Table 1 does not show P values. Therefore, this text is not supported in said table. Add the probability in the text.

Table 2 and 3. Change from N. cerana to N. ceranae and review it throughout the document.

Line 230. Delete point

Lines 259 to 2564. Repeated with the content of lines 90 to 94

I think the discussion should be improved, as much as the results are presented. I understand that this is a case study, however, it is still a discussion of a scientific document, so an attempt should be made to provide an explanation for the results found based on previous research. As well as contrast results. No clear conclusion is reached based on the results obtained.

Line 318. They have no evidence to say that beekeepers are affected mentally and emotionally. Please delete it.

I believe that the language that should be used in a document of this type should be scientific in nature and the use of phrases such as dear bees should be omitted.

Line 336-337 says: Amitraz, and there is 336 evidence of genetic resistance developing in mite populations. This should be supported by references.

Author Response

Referee Number 2:

Comments and Suggestions for Authors

Appreciable authors

I value your work, I consider that your results are important for beekeeping. However, I believe that the presentation of the document should be improved. Below I list some suggestions.

Line 59. The titles should follow the same format, this title is posed as a question although it does not have the question mark. I suggest just putting: sample collection

Done

Line 90. The title does not have the numbering. Add or remove all subtitles

Done

Line 95. Add corresponding numbering, Start with a period

Done

170-171. Table 1 does not show P values. Therefore, this text is not supported in said table. Add the probability in the text.

Table 1 does not have Pvalues because it is reporting descriptive stats only.

Table 2 and 3. Change from N. cerana to N. ceranae and review it throughout the document.

done

Line 230. Delete point

done

Lines 259 to 2564. Repeated with the content of lines 90 to 94

Respectfully, there seems to be confusion with some line numbering between referee’s and what we submitted. The lines you pointed out are not redundant. One set introduces the case study while the other briefly starts the discussion. We are leaving them as is, since it creates continuity in the manuscript. However, in the process of going over this I did find a large redundancy and addressed that. 246 – 260 that was a restatement from results.

I think the discussion should be improved, as much as the results are presented. I understand that this is a case study, however, it is still a discussion of a scientific document, so an attempt should be made to provide an explanation for the results found based on previous research. As well as contrast results. No clear conclusion is reached based on the results obtained.

Hi, I went through the discussion line for line after reading this comment. As you acknowledged this is a case study. We talked as authors and will not try to draw more conclusions or explanations than are appropriate at this time. We have key citations that link our observations to already established work. See below. However, we believe extending our discussion could serve as a disservice to stakeholders since synergisms, nutritional stress and pesticides were not investigated.

Symptoms and losses were reflective of national losses experienced in 2007-2009[10-12].

Detected at high levels in many colonies, N. ceranae has been previously implicated in sudden colony losses[14]. 

Persistent, high levels of infestation like this create concern over the health of operations going forward as infested colonies can re-infest neighboring colonies through the dispersal of mites while simultaneously promoting ongoing viral circulation within operations[12, 13].

Line 318. They have no evidence to say that beekeepers are affected mentally and emotionally. Please delete it.

The anonymous responses we provide from interviews are proof of mental and emotional stress. As stated in our methods we were including them in a way that would protect anonymity. We placed these in lines 290 – 302.

I believe that the language that should be used in a document of this type should be scientific in nature and the use of phrases such as dear bees should be omitted.

I searched for the word “dear”. It does not exist in the manuscript. So I don’t know what you are referring to. This is a scientific document, and it is one that is written in a way that provides important information to researchers while also readable to stakeholders and policy makers.

Line 336-337 says: Amitraz, and there is 336 evidence of genetic resistance developing in mite populations. This should be supported by references.

The reference was added.